# Location of Latent Forensic Traces Using Multispectral Bands

**DOI:** 10.3390/s22239142

**Published:** 2022-11-25

**Authors:** Samuel Miralles-Mosquera, Bernardo Alarcos, Alfredo Gardel

**Affiliations:** 1Police—Specialist of Forensic Image of General Headquarters of Forensics Police, National Police, c/Julián González Segador s/n, 28043 Madrid, Spain; 2Polytechnic School, University of Alcala, 28805 Alcala de Henares, Spain

**Keywords:** forensic science, photography, multispectral, latent traces, blood, ultraviolet, infrared

## Abstract

In this paper, a conventional camera modified to capture multispectral images, has been used to locate latent forensic traces with a smart combination of wavelength filters, capturing angle, and illumination sources. There are commercial multispectral capture devices adapted to the specific tasks of the police, but due to their high cost and operation not well adapted to the field work in a crime scene, they are not currently used by forensic units. In our work, we have used a digital SLR camera modified to obtain a nominal sensitivity beyond the visible spectrum. The goal is to obtain forensic evidences from a crime scene using the multispectral camera by an expert in the field knowing which wavelength filters and correct illumination sources should be used, making visible latent evidences hidden from the human-eye. In this paper, we show a procedure to retrieve from latent forensic traces, showing the validity of the system in different real cases (blood stains, hidden/erased tattoos, unlocking patterns on mobile devices). This work opens the possibility of applying multispectral inspections in the forensic field specially for operational units for the location of latent through non-invasive optical procedures.

## 1. Introduction

If there is an axiom that has gone along with photography since its creation, it is the advertising slogan from the *Ermanox* camera, (the camera used by the pioneering photojournalists of the early 20th century) promising the capture of all kinds of indiscreet snapshots, even in impossible lighting conditions: *“with Ermanox, my eye becomes my exposure meter, if I can see it, I can photograph it!”* [1]. By its nature and intended purpose, photography has always been traditionally related to the capture of images within the visible light spectrum, trying to imitate with it the strict patterns of perception of the human eye. However, the demands of a discipline, such as forensic photography, have made it necessary to progressively incorporate new techniques, in many cases far away from the so-called conventional photography. The advances in tools and methods have taken forensic photography far beyond the limited human perception, being especially significant multispectral technology that, despite the associated problems of aliasing [2,3] and artifacts derived from image conditions [4] as a non-invasive procedure for locating vestiges, it has allowed new fields of research to open [5] and minimizing the risks of deterioration or accidental deletion of latent traces, a common problem of classical inspection procedures [6]. There are multiple related fields where multispectral image analysis has been used, for example, to detect certain chemicals in food [7] to detect dangerous gas emissions [8] and, in forensic sciences to retrieve shooting distances [9]. Remote sensing is the main research field where multiple bands of information is considered, thus several advances in hyperspectral images related to reduce band interference and select the best band has been proposed [10]. In our paper, we focus the research work on the forensic inspection to retrieve latent vestiges, considering the advantage of the method because the risks of deterioration or accidental deletion of traces is reduced compared with traditional inspection procedures [11].

The continuous evolution of optical image capture devices, in any of its variants, has contributed, not without some initial scepticism, to the definitive incorporation of this technology into the field of Forensic Sciences, being today an essential tool for any forensic researcher. This progress, especially in the last 45 years with the emergence of electronics and computing in the photography sector, has not only increased the performance of this technology but also improving the sensitivity, precision and definition of current optical devices [12,13]. In addition, the appearance of new techniques and capture image devices has led to greater and faster access to the latent information to be obtained, regardless of whether or not they are initially visible.

Every investigator looking into a crime scene to locate those vestiges relevant to the clarification of a case, must use, during the forensic inspection of the scene, each and every one of the available detection procedures and tools. Therefore, from the definitive incorporation of photography into the field of forensic sciences, one of the main search tools has been the “illumination” of the scene with visible light (VL), which represents only a small fraction of the electromagnetic spectrum contained in the 380–740 nm wavelength band, corresponding to a frequency band of about 405–790 THz. Radiations with wavelengths bordering the limits of the VL, especially ultraviolet (UV) and infrared (IR) radiation, are also used in criminalistics for the detection and photographing of vestiges, both visible and latent, thanks to the incorporation of forensic lights, named as alternate lighting sources (ALS) which use tunable wavelengths for illumination of the scene [14]. A large part of the ALS radiated spectrum is not perceptible by the human eye, being outside the spectral curve of luminous efficiency of its photoreceptors, but it is captured by certain photosensitive emulsions used in old argentic/analogue photography or, currently, multispectral/hyperspectral capture devices. These optical localization procedures provide the researcher with several advantages [15]. In addition to providing more information about the scene than narrow-band visible sensors [16]. On the one hand, as a non-contact inspection method there is no risk of deterioration of traces nor cross-contamination of the scene itself. On the other hand, the inspection results can be evaluated instantaneously, reducing the processing times and permanence of the researchers at the scene, improving the performance and moving the searching process in the right direction. Finally, it is worth noting that the optical inspection can be complemented with other traditional procedures to locate latent vestiges by means of physical developers or chemical components, such as blood chemo locators, Luminol, Bluestar, powdery developers of sweat and sebaceous secretions, black magnetic, wax or fluorescent powder.

Just as the resolution of a digital image is determined by the number of pixels of its sensor, the visual perception that the human being has of its environment is given, at first, by the density of photoreceptors located in its retina, called cones and rods for the characteristic shape of its external segments. Its main function is to absorb light and, thanks to the visual pigments they contain—proteins such as opsin and rhodopsin—transform it into electricity, which is the only form of energy that can be processed by the brain. The cones are responsible for the so-called photopic or diurnal vision, while the rods on the other hand are more sensitive (scotopic or nocturnal vision conditions) but their response to light is more moderate, which translates into lower visual acuity and reduced discrimination of colour. Figure 1 shows a comparison of the spectral curve of luminous efficiency of the cones and rods of the human retina, whether under conditions of photopic or scotopic vision, with those of a conventional digital camera with a CMOS silicon sensor. It can be observed that the nominal spectrum sensitivity of the camera is considerably higher than the human-eye photoreceptors, specifically from about 380 to 740 nm versus 300 to 1150 nm, both at short wavelengths (UV radiation) and at long wavelengths (IR radiation).

To avoid certain optical aberrations associated with the radiations of the non-visible spectrum, that is, those that are below 380 and above 740 nm wavelengths, the manufacturers of cameras place in front of the silicon sensor a colour filter array (CFA) and a bandpass filter, generically called hot mirror, whose mission is to block all UV and IR radiation, allowing only VL wavelengths to reach it. The function of the *hot mirror* is to limit the spectral sensitivity of the sensor to adapt it to human perception, eliminating possible optical aberrations associated with these specific regions of the spectrum, so that the images captured by the camera are the same as those perceived by the photographer. On the other hand, Forensic Photography requires that the optical devices capture regions of the spectrum outside the strict human perception, since it is often there where the reactions of the electromagnetic radiation with the relevant forensic vestiges occur and their observation is critical for the subsequent clarification of the fact to be investigated.

Generically, multispectral imaging refers to the capture of images using more than one spectral band, regardless of whether they are contiguous and whether they are visible to the human eye in which the result is processed as an individual image. Therefore, multispectral imaging uses a subset of specific wavelengths within a defined spectral range. The concept of multispectral imaging has its origins in the mapping of the Earth’s surface using artificial satellites, such as the pioneer LandSat-1 launched in 1972 and equipped with a quad-band multispectral scanner system. As this technology became more affordable, it was taking more and more presence, expanding its uses to the digitization of cultural goods in the 1980s, and in recent years, to the location of vestiges in the field of Criminalistics [17].

Currently, the multispectral capture devices available in the market, incorporate different types of sensors, depending on the model and the specific nominal sensitivity to the spectrum that is necessary according to the demands of the addressed work.

On the one hand, there would be the sensors InGaAs, indium gallium arsenide, used in infrared cameras and which have, depending on the manufacturer, a spectral curve of luminous efficiency wavelengths of 700 to 1.700 nm. These devices are commonly used in the inspection of industrial phenomena but, from a strictly forensic point of view, their sensors, although they are useful in the detection of certain latent evidence that reacts to long wavelengths. In the cases of blood, alterations in documents or shooting residues, the captured images do not have sufficient quality to provide the vestige with the necessary identifying value, with a general lack of sharpness and contrast that reduces its effectiveness [18].

Another type of sensors would be the so-called Vidicom tubes which, in essence, are video cameras with a lead oxysulfide sensor. They are capture devices commonly used in examinations of artworks, in particular paintings on canvas. Its spectral curve of luminous efficiency is variable, being able to reach up to 1.900 nm wavelength. Its application in Criminalistics has several drawbacks, derived largely from its low resolution, which ultimately translates into a difficulty in reproducing elements or details of size less than 2 mm. In addition, the captured images present large geometric aberrations that make it impossible to obtain measurements of elements within the image. Finally, the reduced field of view forces to work with image mosaics so their use during a police technical inspection is very scarce [19].

Thermal imaging sensors based on uncooled microbolometers, or lead selenide or indium antimonide, offer a nominal spectral sensitivity of 1.500 nm and higher wavelengths [20]. The main limitation presented by this type of sensors for forensic use is that the main vestiges of criminalistic relevance do not emit any thermal reaction, beyond the heat generated by the action of cadaveric fauna.

In the same way that there has been a natural evolution of the traditional systems of capturing images in monochrome bands to those others in colour bands, already assuming a considerable improvement in the quality and usefulness of them, also the capture devices have evolved from capturing only the visible spectrum to those capable of capturing images in multispectral bands not visible to human-eye. The incorporation of these devices into the field of criminalistics has had a direct utility as a tool to detect elements not perceived with the naked eye, through a direct comparison with the object to be inspected or with the visible image of it, visually analysing the differences between the two. In forensic sciences, the usefulness of these multispectral devices lies in the fact that they allow to visualize and document the reactions that, in the face of electromagnetic radiation of a specific wavelength, present certain components of both the vestiges to be located and the surface where they settle.

The multispectral capture systems used today in criminalistics for the search for latent vestiges, offer the possibility of locating and photographing evidence in multiple scenarios and work circumstances. These tools are presented in the form of portable tablets with limited functionality, such as the ForenScope tool, intended for fieldwork during the police technical inspection at the scene or as independent workstations, as is the case of foster and Freeman’s DCS 5, oriented exclusively to the work of localization and documentation of pieces of evidence in a forensic laboratory [21]. Both proposals incorporate autonomous filtering and lighting systems in the form of LED rings located around the optics that emit, according to the vestige to be located, different wavelengths, but without the possibility of varying their angle of incidence on the search surface. However, reasons, such as their high cost or a functionality often poorly adapted to fieldwork, mean that they are currently not used by many criminalistics sections. These are the reasons why the Scientific Police in Spain use a low-budget multispectral modified digital camera, without reducing its operability and effectiveness when it comes to locating latent vestiges in a crime scene.

The modified camera, together with a quartz or fluorite lens and an adequate longpass or shortpass filter, would allow to locate and graphically document certain latent traces/vestiges, such as latent blood stains, tattoos partially erased or covered under new tattoos and unlock patterns on mobile phones, among others. Currently, such vestiges are located by physical developers and chemical reagents for forensic use, such as Luminol, Bluestar, Hemascein or Benzidine in the case of latent blood; or magnetic, wax, fluorescent powder, cyanoacrylate or diazafluorenone plus flavin, in the case of traces of sebaceous and sudoriparous secretions.

These classic localization procedures have a direct impact in the scene as they require direct contact with the surface on which the vestige might be present. This fact means that, when applied, either by means of sprayers or fiberglass brushes, they do not allow the analysis of the trace’s morphology nor the subsequent search for other latent vestiges, such as fingerprints. This problem is solved using a non-contact localization procedure, such as the modified multispectral modular system presented in this paper.

The rest of the paper is organized as follows. Next section presents the materials and methodology used in the research, to show the correct procedure to retrieve latent forensic traces. Section 3 presents the conducted experiments, resume tables and detection results for real use cases (blood stains, hidden/erased tattoos, unlocking patterns on mobile devices, etc.) showing the validity of the proposal. Finally, Section 4 resume the main conclusions of the research work.

## 2. Materials and Methods

Electromagnetic radiation can interact with the matter through an exchange of energy, being able to classify this phenomenon according to several criteria, such as nature of the matter involved, specific incident spectral band or the possible reactions of the interaction. Starting from the fact that the laws of spectrophotometry were enunciated based on monochromatic electromagnetic radiation acting on a homogeneous system, when electromagnetic radiation affects a certain material, we understand that, for the purposes of this study, a vestige or the surface where it sits, the beam will produce different reactions depending on the energy levels in its atoms, it may be absorbed, transmitted, dispersed, reflected or it may induce photoluminescence [22].

When the atoms of a certain material are irradiated with electromagnetic radiation, part of that form of energy can be absorbed by the atoms of the material which, as a consequence, will pass from a lower energy state (basal state or *E*_1_) to a state of higher energy (excited state or *E*_2_). For this absorbance reaction to occur, the energy of the photons of the incident beam (*h · v)*, must be equal to the energy difference between *E*_1_ of the atoms of the material and posterior energy *E*_2_.
Δ*E* = *E*_2_ − *E*_1_ = *h* · *v*(1)

The atoms of the material can return to their *E*_1_ by converting the energy of their *E*_2_ either into heat, through a luminescent reaction or through a photochemical reaction, which can be documented, by means of the appropriate capture devices, evidenced by a darkening of the irradiated material for the duration of the induced reaction. The transmission of electromagnetic radiation assumes that the beam that is not absorbed or reflected, will pass through the sample without suffering perceptible or energetic changes. As the wavelength of the incident radiant flux increases, the absorption of radiation by the sample decreases, increasing the transmittance, so that the energy absorbed and reflected is less than that transmitted, which will be evidenced in a partial disappearance of the element in the captured image. Dispersion occurs when the incident beam is absorbed and immediately emitted uniformly in all directions, without any energetic change. A reaction that occurs when radiation hits particles in the sample that are smaller than the wavelength of the incident beam itself, so they polarize and oscillate at the same frequency as the radiation, acting as a source that propagates in all directions. Photoluminescence involves an excitation of the particles that make up the sample by absorbing the incident radiation, emitting fluorescence, as happens in scattering, in all directions.

The relevance of these phenomena, for the purposes of this study, is that a large part of the relevant forensic vestiges for the possible clarification of facts of police interest and on which the subsequent expert and technical reports in each of the forensic disciplines will be based, present distinct reactions, visible or not to the human eye, depending on the wavelengths of spectrum used to induce them and the specific region of the spectrum where they occur. This feature leads to a characteristic behaviour that can be used, as a spectral fingerprint, to achieve its location and subsequent graphic documentation [23].

In this paper for the development of the multispectral camera it has been used a Nikon camera D3500-24.2 megapixels, reflex CMOS APS-C 23.5 × 15.6 mm, commonly available in the Scientific Police units in Spain and that meets the needs of the fieldwork of a crime scene investigator: ease of handling the different modules of the multispectral system, ability to view the scene before capturing, interchangeable bayonet to mount quartz optics e.g., Nikon UV. Reusing the camera currently owned by operational units significantly reduce the cost, making available the multispectral technology. The multispectral techniques shown in the paper has also been tested with other cameras/sensors capturing multispectral images such as smartphones and action cameras, but its operation, focus range and resolution is reduced when compared with digital SLR cameras. Figure 2 shows the modified Nikon D3500 camera. The bandpass *hot mirror* filter—CFA colour filter matrix—has been extracted and replaced by a full spectrum filter, to increase its nominal sensitivity [24]. The sensor has been conveniently shifted closer to the nodal point of the lens, recalibrating it in order to correct the image focus, important when it is necessary to locate latent traces using infrared illumination. Likewise, for those images in which UV radiation is used, the camera lens has been replaced by a quartz or fluorite lenses, both permeable to UV radiation, because the Crown glass used by commercial photographic lenses blocks short wavelengths. This conversion allows the capture of regions of the UV and IR spectrum, losing colour information.

For the analysis of pieces of evidence presented in the next section, different bandpass and shortpass filters have been used when filtering the specific wavelengths of the lighting sources and, thus, making visible the reactions of the latent traces. We have carried out a field study on the effectiveness of multiple long-pass and shortpass filters from different manufacturers to detect latent traces in multiple situations. The working band of the used filters (Wratten 87, Schott RG9 and Baader UV/IR-cut) are shown in Figure 3 where it can be shown their spectral response to the electromagnetic radiation used to locate the latent vestiges.

We have named the modular multispectral inspection system as Invespector being composed by the modified camera, optical lenses, band filter and the illumination source. The right combination of the different elements leads to an effective retrieval of latent traces.

## 3. Results and Discussion

This section presents the results obtained using the Invespector capturing device for the localization of multiple and diverse pieces of evidence: latent blood stains, erased tattoos and unlocking pattern of mobile devices. The source of data comes from real cases analysed by the Spanish Scientific Police.

### 3.1. Localization of Spots and Latent Blood Patterns

One of the applications in which the effectiveness of the modified multispectral camera has been tested is in the location and photographing of latent blood. Blood stains and patterns are one of the most common vestiges present at any scene of a violent crime. Whether during the police technical inspection conducted at the scene or later in the forensic laboratory, it always acquires special relevance to locate and document this type of biological vestiges because, in addition to being a source of genetic and serological material, they might provide significant information for blood pattern analysts, allowing them to sometimes reconstruct the very commission of the fact, by studying the morphology, size and location of blood stains in the scene or on the objects located in it.

When the blood is recent, has not been manipulated and is settled on surfaces that contrast with the sample itself, its location and photography is a simple task. Unfortunately, it is common for blood stains to be difficult to detect, either because they are settled on dark surfaces that mask them, or because they have been intentionally cleaned or partially erased as a result of the passage of time. It will be then when the researcher uses presumptive methods of locating latent blood, both with chemical and optical means [25]. Blood, human or animal, contains certain components such as hemoglobin iron, lipids and proteins that react through intense absorbance when illuminated with wavelengths typical of ALS forensic lights in its spectrum and near IR wave radiation [26]. This reaction makes the blood vestige latent even in low dilutions (1:200) [27] or small amounts darken, which facilitates its detection and subsequent photographing, provided that the surface where it sits reacts by reflecting or transmitting those specific wavelengths used to induce the reaction [28].

Figure 4 shows the detection results for a latent blood settled in a black cotton shirt. The classical visible images were compared with the image captured using a multispectral Invespector device with a source of IR radiation of 850 nm, a capture angle of 90° and attaching to the camera lens an IR longpass filter Schott RG9. As it can be seen, it is possible not only to locate and document latent blood settled in a black cotton shirt, but to respect the morphology of the stain itself, in this case a footstep. Additionally, the cotton shirt is treated with the Bluestar chemo locator which can react with blood in dilutions of up to 1:10,000.

Although optical procedures of localization in multispectral bands are less sensitive to the presence of blood than the classic chemical procedures, optical procedures offer several advantages:It is a non-invasive procedure that is with no contact, so it is possible to conduct subsequent comparative morphological studies.It preserves the morphology of the latent blood stain, for example, in this case, the footprint of the footwear. It can be shown that only Figure 4b can be correlated with the footprint provided in Figure 5.


It does not interfere with the subsequent DNA study of the localized sample.It allows the subsequent search for lophoscopic traces if the surface is adequate.The high penetration capacity of IR radiation on certain surfaces. An example is the location of blood settled under layers of up to 5 mm of acrylic paint, wallpaper or cotton or nylon fabrics, as it can be seen in the images of the Figure 6.


Additionally, in order to locate latent blood, UV radiation of 365 nm wavelength has been shown to be equally effective [29]. The two images that make up Figure 7 correspond to a fragment of cotton sheet from the scene of a violent crime. In this case, despite the fact that the sheet had been washed, using the multispectral device it was possible to locate remains of what a laboratory later certified to be human blood. To do this, the effect was illuminated with UV radiation of 365 nm with an angle of incidence of 90°, attaching to the camera a fluorite lens that provides permeability to the UV radiations and a UV shortpass filter Baader UV/IR-cut. Under these circumstances, and despite the fact that the amount of vestige was very small, after the sheet was subjected to washing, the blood vestige could be detected without the need to apply chemical locators for it.

We have run multiple tests to show the capabilities of the multispectral device. Table 1 shows the right configurations for the detection of a drop of blood on different surfaces. For example, to search for blood stains in a denim-fabric, the best configuration is a Crown-glass lens with a Hoya R72 filter, illuminating the scene with a 720 nm UV source, capturing the sample in vertical (90°).

### 3.2. Detection of the Unlocking Pattern of Mobile Devices

Another application in which the Invespector multispectral device has been shown to be effective is in the detection of unlocking patterns of mobile phones not visible to the human-eye. As it can be seen in Figure 8, in this case the unlocking pattern could be detected, illuminating the phone with IR radiation of 950 nm wavelength at an angle of incidence of only 10°, attaching to the target of the multispectral device an IR longpass filter Wratten 87.

The sebaceous secretions deposited by repetition of the movement on the screen of the mobile phone provoke the absorbance reaction to the IR radiation of 950 nm wavelength. Additionally, the outer layer of Indium Tin Oxide (ITO) [29] that coats the glass of the phone screen, as a consequence from the repeated friction of the user in the area of the unlocking pattern, modifies the reflectance reaction of the glass that is below the ITO coating making it more visible in the captured image. The Invespector device captures the contrast between the absorbance reaction of the sebaceous secretions and the ITO layer; and the reflectance of the glass below the ITO. Therefore, it is possible to recover the repeated finger movement to input a similar pattern and unlock the smartphone.

### 3.3. Locating Covert or Partially Erased Tattoos

Tattoos, like any other birth mark or supervening mark (wounds or scars), have been used as an authentication particular mark complementary to other identifying elements of the individual/person. In 1853, a court in Paris commissioned the pathologist Auguste Tardieu a study on the indelible condition of tattoo inks, the medico-legal interest in tattooing as a sign of identity capable of providing relevant information about its wearer gradually grew until it was incorporated, like any other distinctive physical characteristic, to the written and photographic police portrait of Alphonse Bertillon. However, as Tardieu claimed in his forensic study, *“despite the permanent character of the tattoo, they can always be made less noticeable, altered or increased by time”* [30].

Most of the pigments that make up the inks used in tattoos are derived from charcoal (black ink) or metals such as iron (black ink), mercury (red ink), cobalt (blue ink), chromium (green ink), titanium/zinc oxide (white ink) or cadmium (yellow ink) [31]. These substances have the property of reacting by transmitting or absorbing, to a greater or lesser extent, the wavelengths of the IR spectrum up to 1050 nm [32], which makes the tattoo either disappear or darken and become more visible, by increasing its contrast with human skin, which in turn reacts by reflecting NIR (Near Infrared) radiation.

Figure 9 shows how the multispectral camera coupled with an IR longpass filter Wratten 87 is capable to capture the transmittance and absorbance reactions of the different components of the inks used in the tattoo. In this particular case, this individual has hidden a small latent tattoo, with the now readable text “caliente”, under another tattoo of the shield of the “Valencia CF”. Although the ink is injected between 1.5 and 2 mm under the epidermis, the high penetration capacity of IR radiation −850 nm wavelength at an angle of incidence of 90°—cause certain pigments that make up the inks of this second tattoo—cobalt in the blue ink, cadmium in the yellow, chromium in the red—react by transmitting/reflecting the IR radiation, so they partially disappear in the final image. This make the underlying tattoo perceptible as its ink composition—iron of the black ink—reacts by absorbing the IR radiation, that is, darkening the text.

As a conclusion, it can be said that this technique is highly effective to visualize hidden elements, partially erased or covered under external layers, when the elements located in these upper layers react according to their own composition modifying the transmittance/absorption of current spectrum illumination.

Multiple tests and combinations of the filter and illumination modules have been done to show the capabilities of the multispectral device to detect skin tattoos. We have concluded that, in the case of eternal ink, the best option is to use a crown glass lens, a Wratten 87 filter, illuminate the scene with an IR source (850 nm) and capture the image with a 90° angle. Table 2 shows the ink’s reaction for the infrared radiation based on the components for each ink colour.

## 4. Conclusions

In this paper, an effective procedure has been shown to detect latent vestiges by modifying a conventional camera, from those assigned to the scientific police units in Spain to its multispectral version, thus increasing its curve of luminous efficiency and nominal sensitivity.

Several processing results are shown to demonstrate the efficiency of such a system considering not only the multispectral camera but different ways of filtering the light captured from IR/UV illumination considering both the radiation reflected/absorbed by the scene or vestige.

The effectiveness of this multispectral procedure has been shown in the location and graphic documentation of latent vestiges present in the crime scene itself, such as traces of stains or blood patterns, and in the pieces of evidence to be analysed in the laboratory, such as the detection of hidden or partially erased tattoos, or the visualization of unlocking patterns on mobile devices. Similar procedures can be carried out either during the technical inspection of the crime scene itself or later in the forensic laboratory.

Currently in forensic sciences, the usefulness of multispectral devices on the market lies in the fact that they allow to visualize the reactions that, in the face of electromagnetic radiation, present certain intrinsic and extrinsic components of these vestiges to be located. However, its high cost and an often poorly adaptation to field work reduces its use. The modified digital SLR camera, with a nominal sensitivity beyond the visible spectrum, open the possibility of incorporating this technology to police departments that currently lack it, complementing these new non-invasive procedures for locating latent vestiges.

## Figures and Tables

**Figure 1 sensors-22-09142-f001:**
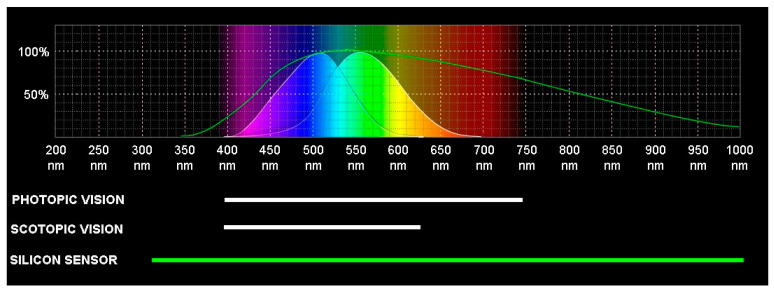
Normalized spectrum sensitivity for photopic and scotopic vision of the cones and rods of a human eye compared with a CMOS sensor.

**Figure 2 sensors-22-09142-f002:**
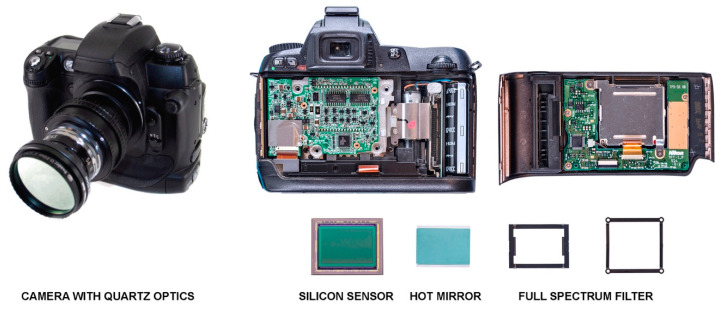
Multispectral capture device modified with quartz optics and Baader UV/IR-cut coupled.

**Figure 3 sensors-22-09142-f003:**
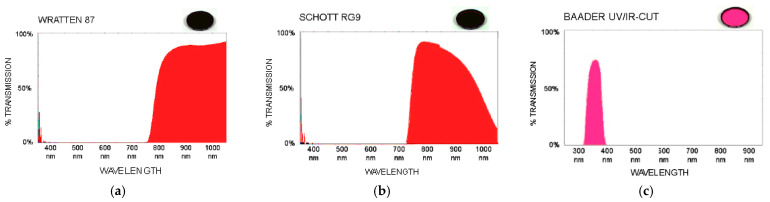
(**a**) IR Longpass filter Wratten 87; (**b**) IR Longpass filter Schott RG 9; (**c**) UV Shortpass filter Baader UV/IR-cut.

**Figure 4 sensors-22-09142-f004:**
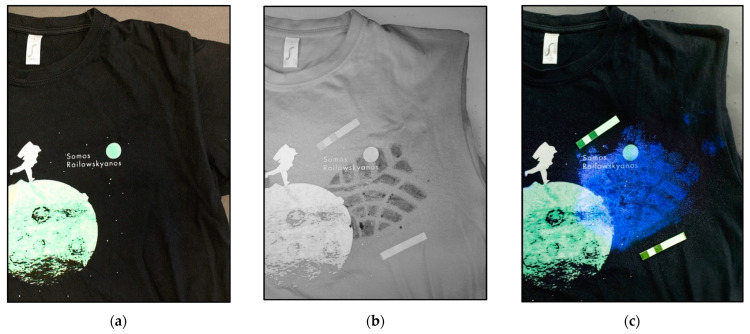
Images of black cotton shirt with traces of latent blood photographed (**a**) with conventional camera and illuminated with white visible light, (**b**) by Invespector illuminated with IR radiation 850 nm and (**c**) after applying a chemo locator, such as Bluestar.

**Figure 5 sensors-22-09142-f005:**
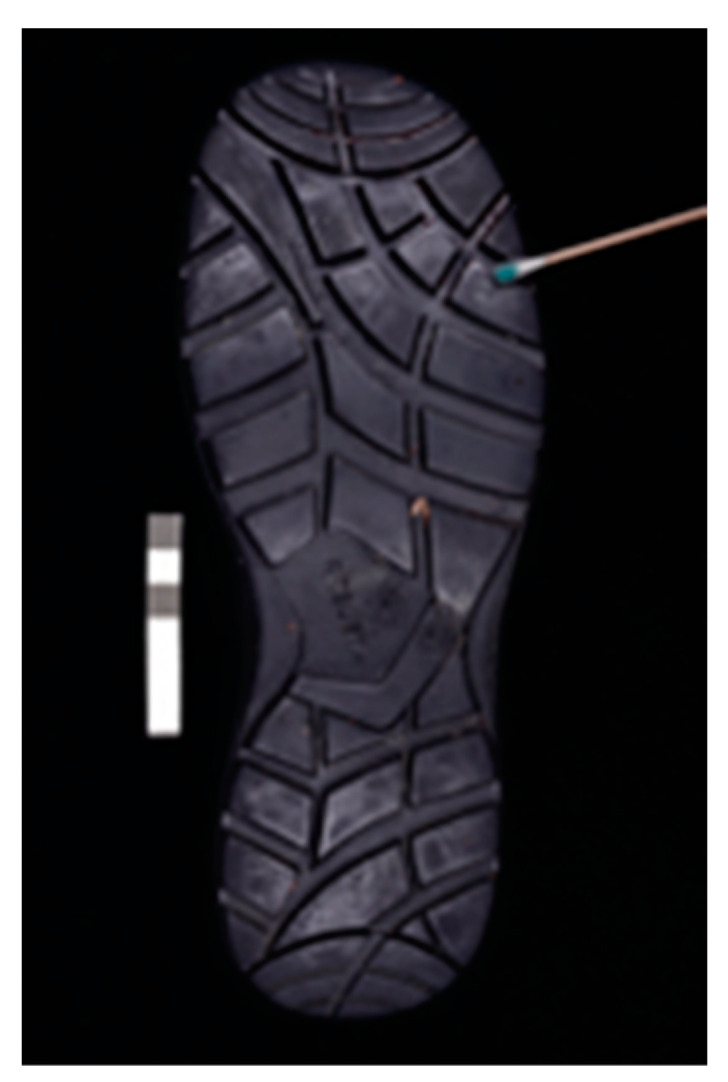
Footprint of the shoe used in the T-shirt under inspection.

**Figure 6 sensors-22-09142-f006:**
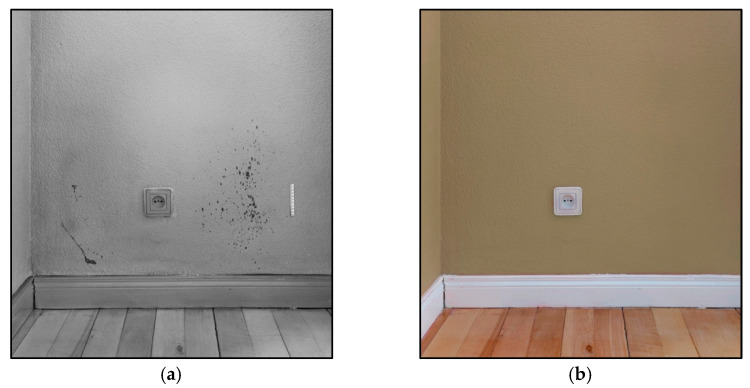
Images of a wall with traces of latent blood under a layer of acrylic paint: (**a**) photographed with a conventional camera and illuminated with white visible light and (**b**) by Invespector illuminated with IR radiation 850 nm.

**Figure 7 sensors-22-09142-f007:**
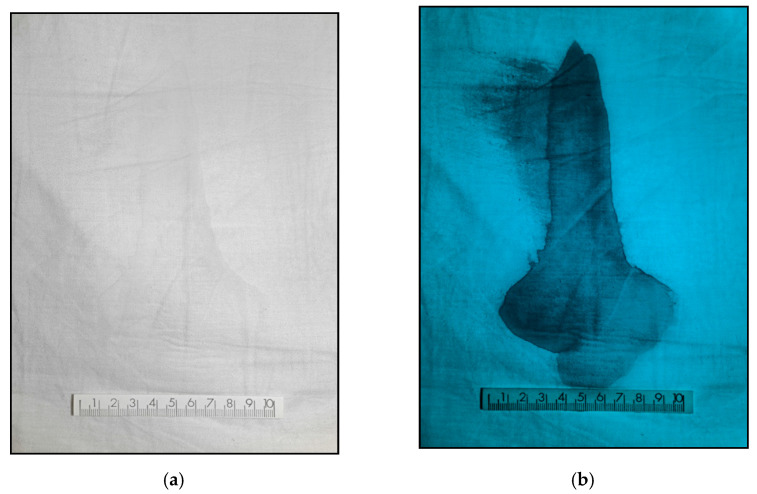
Images of a cotton sheet with traces of latent blood photographed: (**a**) with a conventional camera illuminated with white visible light and (**b**) Invespector system using UV illumination 365 nm.

**Figure 8 sensors-22-09142-f008:**
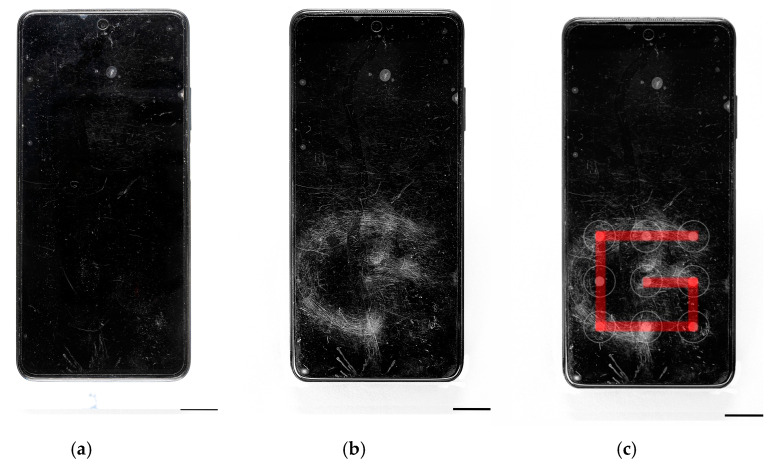
Images from a mobile phone photographed (**a**) with a conventional camera and illuminated with white visible light; (**b**) captured by Invespector camera and illuminated with IR radiation 950 nm; and (**c**) the pattern movement recovered to unlock the smartphone successfully.

**Figure 9 sensors-22-09142-f009:**
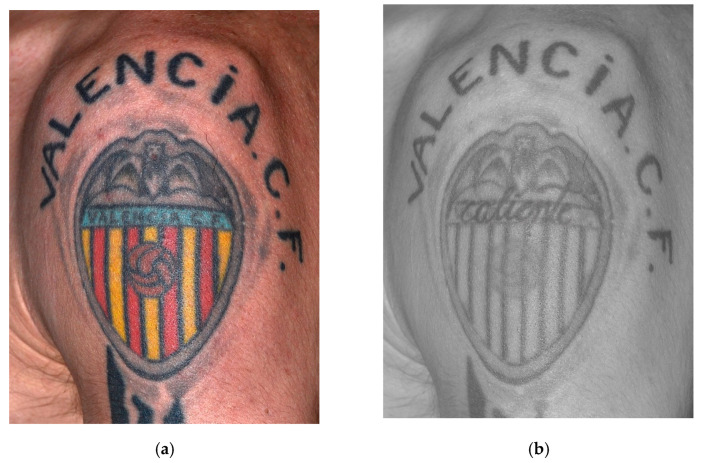
Images of tattooed shoulder: (**a**) captured with a conventional camera and illuminated with white visible light, and (**b**) captured by Invespector and illuminated with IR radiation 850 nm.

**Table 1 sensors-22-09142-t001:** Test results for the detection of a drop of human blood -0.05 ml- considering different surfaces.

Surface Configuration	On Steel	On Washed Cotton	On Wood	On Tile	On Denim Fabric	On ABS Plastic	On Cotton	Under Acrilic Paint	On Nylon
**Lens type**	quartz	quartz	Crown glass	Crown glass	Crown glass	Crown glass	Crown glass	Crown glass	Crown glass
**Filter type**	Kolari UV	Baader UVIR	none	none	Hoya R72	Hoya R72	Schott RG9	Schott RG9	Wratten 87
**Illumination**	365 nm	365 nm	415 nm	455 nm	720 nm	750 nm	850 nm	850 nm	950 nm
**Capturing angle**	25°	90°	45°	45°	90°	45°	45°	90°	45°
**Surface’s reaction**	reflectance	reflectance	reflectance	reflectance	transmittance	absorbance	reflectance	transmittance	transmittance

**Table 2 sensors-22-09142-t002:** Reactions of different tattoo inks to infrared radiation.

Tattoo Inks	Black	Red	Blue	Green	White	Yellow	Orange
**components**	Iron/charcoal	Mercury	cobalt	chromium	zinc oxide/titanium	cadmium	Iron/cadmium
**Ink’s reaction**	absorbance	reflectance	transmittance	transmittance	reflectance	reflectance	absorbance

## Data Availability

Not applicable.

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
