# Peer review of "Location of Latent Forensic Traces Using Multispectral Bands"

_sensors, 2022, doi:10.3390/s22239142_

Round 1

Reviewer 1 Report

Multispectral image recognition is important research in the field of computer vision. To realize the location and graphic documentation of latent vestiges present in the crime scene itself, the authors presented an improvement. Although this paper is interesting, I still have the following concerns:

1. You describe the origin of the problem in your abstract. You don't talk about the problem you want to solve or the methods you use. It makes me wonder what your motives are.

2. In the Introduction, the author uses a lot of space to introduce the origin of the camera, which has nothing to do with the problem to be solved in this paper.

3. The authors seem to credit one innovation: The bandpass hot mirror filter-CFA color filter matrix- has been extracted and replaced by a full spectrum filter, to increase its nominal sensitivity. But this is existing work, and the authors themselves have given references. I'm curious about what innovations or modifications you've made yourself and what your motivations are.

4. The experiments were sorely lacking, and I did not see the performance of the comparative experiments. I don't know the source of the data you're using, so I question the point of this work.

5. Multispectral image recognition is a flourishing task. However, the authors did not review and discuss the previous work, and the paper was not theoretical enough. Here are some related works that may help your work:

1)Mosaic Convolution-Attention Network for Demosaicing Multispectral Filter Array ImagesIEEE TCI.2021.

2)Discriminative Suprasphere Embedding for Fine-Grained Visual Categorization. IEEE TNNLS, 2022.

3)Demosaicking Using a Spatial Reference Image for an Anti-Aliasing Multispectral Filter ArrayIEEE TIP.2019.

Author Response

Dear Reviewer:

We would like to thank you for your comments helping us to improve the paper. Please receive a modified copy of our paper. New contents are highlighted with track changes activated.

We have taken all your suggestions into account, below you can find our comments.

Reviewer comments

Multispectral image recognition is important research in the field of computer vision. To realize the location and graphic documentation of latent vestiges present in the crime scene itself, the authors presented an improvement. Although this paper is interesting, I still have the following concerns:

  1. You describe the origin of the problem in your abstract. You don't talk about the problem you want to solve or the methods you use. It makes me wonder what your motives are. In the Introduction, the author uses a lot of space to introduce the origin of the camera, which has nothing to do with the problem to be solved in this paper.

Following the reviewer’s suggestions, we have improved the abstract to clearly show the paper contributions, obtaining latent forensic traces from a crime scene using a modified digital camera attached to the adequate optic lens and wavelength filters with the suitable illumination to capture an image where the traces are shown in the best possible way.

We have also modified the introduction providing the reader with the paper motivation, the problem to be solved and reducing the contents related to the origin of multispectral cameras as suggested by the reviewer.

  1. The authors seem to credit one innovation: The bandpass hot mirror filter-CFA color filter matrix- has been extracted and replaced by a full spectrum filter, to increase its nominal sensitivity. But this is existing work, and the authors themselves have given references. I'm curious about what innovations or modifications you've made yourself and what your motivations are.

We agree with the reviewer that the paper needed to clarify better that the contributions are not related with the modification of the digital camera but the combination of different modules: multispectral camera, lenses, filters, and illumination sources to capture the latent forensic traces.

In recent years, different multispectral capture devices adapted to forensic tasks have been proposed. However, they are meant for high-end forensics laboratories (high cost and non-adapted to field work). The motivation of our work was to develop a multispectral device that can be used by operational forensic units. We have tested its effectiveness locating latent traces in the field with real cases.

Following these comments, we have included text and references to clarify that the paper contributions are related to the use of the non-invasive multispectral system composed of several modules for capturing the hidden latent traces from a crime scene.

  1. The experiments were sorely lacking, and I did not see the performance of the comparative experiments. I don't know the source of the data you're using, so I question the point of this work.

Following the suggestions of the reviewer we have added more information about the experiments conducted to lead to the final detection results for the different use cases shown in the paper. We have added 2 tables, one related to the blood stains detection for different surfaces and another related to the ink colour used in tattoos. The information of the tables contains the resume of a large number of tests using the different modules (lens, filter, and illumination) with the multispectral camera. In the result section we have added that the source of the data comes from real cases analysed by the Spanish Scientific Police.

  1. Multispectral image recognition is a flourishing task. However, the authors did not review and discuss the previous work, and the paper was not theoretical enough. Here are some related works that may help your work:

  • Mosaic Convolution-Attention Network for Demosaicing Multispectral Filter Array Images. IEEE TCI.2021.
  • Discriminative Suprasphere Embedding for Fine-Grained Visual Categorization. IEEE TNNLS, 2022.
  • Demosaicking Using a Spatial Reference Image for an Anti-Aliasing Multispectral Filter Array. IEEE TIP.2019.

We have included more references (+9) in the paper, analysing their work and fixing the content in the state-of-art. In particular we have included the references 1 and 3 suggested by the reviewer.

  1. Maru Kawase, Kazuma Shinoda, Madoka Hasegawa, “Demosaicking Using a Spatial Reference Image for an Anti-Aliasing Multispectral Filter Array”. doi:10.1109/tip.2019.2910392
  2. K. Feng, Y. Zhao, J. C. -W. Chan, S. G. Kong, X. Zhang and B. Wang, "Mosaic Convolution-Attention Network for Demosaicing Multispectral Filter Array Images," in IEEE Transactions on Computational Imaging, vol. 7, pp. 864-878, 2021, doi: 10.1109/TCI.2021.3102052.

Once again, we would like to express our gratitude for the relevant effort done reviewing this paper.

Reviewer 2 Report

In this study, the adaptation of a conventional multispectral camera has been carried out to locate latent forensic traces to use the already digital SLR cameras available in the different sections of the Scientific Police in Spain. In my opinion, the manuscript’s strengths are an effective procedure has been shown to detect latent vestiges by modifying a conventional camera, from those assigned to the Scientific Police units in Spain, to its multispectral version, thus increasing its curve of luminous efficiency and nominal sensitivity. However, some weaknesses should be addressed, especially the introduction and experiment.

Major issues:

1) The introduction is too simple. There are only 24 references in this paper, which is not enough to fully explain the significance of this study. I suggest that the authors combine the introduction with relevant work and supplement relevant literature, such as

[1] Super-Resolution Mapping Based on Spatial-Spectral Correlation for Spectral Imagery [J]. IEEE Transactions on Geoscience and Remote Sensing, 2021, 59(3): 2256-2268.

[2] Target-Constrained Interference-Minimized Band Selection for Hyperspectral Target Detection, IEEE Transactions on Geoscience and Remote Sensing, 2021, 59(7): 6044-6064.

2) In the Section 2, the introduction of the method used in this paper is too simple. The hardware modules added by the authors, such as Nikon camera D3500-24,2 megapixels, reflex CMOS APS-C 23,5 x 15,6 mm, should explain in detail why they were selected, and what is their content structure? Which circuit images should be further given.

3) In the experiment, the authors only give visual results. Can you provide relevant quantitative evaluation indicators? In addition, it is recommended to conduct experimental comparison with similar hardware to highlight the advantages of the designed hardware.

Minor issues:

1) What are the specific contributions of this paper? It is suggested to add at the end of the introduction:

2) There are some grammatical errors in the article, which need further careful proofreading. For example, some images are not clear enough. What is the ordinate of Figure 4?

Author Response

Dear Reviewer:

We would like to thank you for your comments helping us to improve the paper. Please receive a modified copy of our paper. New contents are highlighted with track changes activated.
We have taken all your suggestions into account, below you can find our comments.

Reviewer comments 

In this study, the adaptation of a conventional multispectral camera has been carried out to locate latent forensic traces to use the already digital SLR cameras available in the different sections of the Scientific Police in Spain. 

In my opinion, the manuscript’s strengths are an effective procedure has been shown to detect latent vestiges by modifying a conventional camera, from those assigned to the Scientific Police units in Spain, to its multispectral version, thus increasing its curve of luminous efficiency and nominal sensitivity. 

Thank you for your comments and positive feedback.

However, some weaknesses should be addressed, especially the introduction and experiment.

Major issues:

1) The introduction is too simple. There are only 24 references in this paper, which is not enough to fully explain the significance of this study. I suggest that the authors combine the introduction with relevant work and supplement relevant literature, such as

[1] Super-Resolution Mapping Based on Spatial-Spectral Correlation for Spectral Imagery [J]. IEEE Transactions on Geoscience and Remote Sensing, 2021, 59(3): 2256-2268.

[2] Target-Constrained Interference-Minimized Band Selection for Hyperspectral Target Detection, IEEE Transactions on Geoscience and Remote Sensing, 2021, 59(7): 6044-6064.

Following the reviewer’s suggestions, we have modified the introduction providing the reader with more information clarifying contribution of the research work. We have included more references (up to 33) in the paper, rewriting the state-of-art content. In particular we have included the references suggested by the reviewer (refs 4 and 10).

4.    Peng Wang, Liguo Wang, Gong Zhang. “Super-Resolution Mapping Based on Spatial-Spectral Correlation for Spectral Imagery”. IEEE Transactions on Geoscience and Remote Sensing, 2021, 59(3): 2256-2268.
10.    X. Shang et al., "Target-Constrained Interference-Minimized Band Selection for Hyperspectral Target Detection," in IEEE Transactions on Geoscience and Remote Sensing, vol. 59, no. 7, pp. 6044-6064, July 2021, doi: 10.1109/TGRS.2020.3010826.

2) In the Section 2, the introduction of the method used in this paper is too simple. The hardware modules added by the authors, such as Nikon camera D3500-24,2 megapixels, reflex CMOS APS-C 23,5 x 15,6 mm, should explain in detail why they were selected, and what is their content structure? Which circuit images should be further given.

We have clarified in the paper that the contributions are not related with the modification of the digital camera but the combination of different modules: multispectral camera, lenses, filters, and illumination sources to capture the latent forensic traces.
Following the reviewer’s suggestions, we have added information to section 2 about the selection/requirements of a suitable digital camera. The Nikon camera is the one used by Spanish Scientific Police but any SLR digital camera might be used if it “meets the needs of the fieldwork of a crime scene investigator: ease of handling the different modules of the multispectral system, ability to view the scene before capturing, interchangeable bayonet to mount quartz optics”. 

3) In the experiment, the authors only give visual results. Can you provide relevant quantitative evaluation indicators? In addition, it is recommended to conduct experimental comparison with similar hardware to highlight the advantages of the designed hardware.

Following the suggestions of the reviewer we have added more information about the experiments conducted to lead to the final detection results for the different use cases shown in the paper. 
Thus, we have added 2 tables, one related to the blood stains detection for different surfaces and another related to the ink colour used in tattoos. The information of the tables contains the resume of a large number of tests using the different modules (lens, filter, and illumination) with the multispectral camera. 
Additionally, in section 2, we we state that is possible to use the same procedure with other cameras/sensors capturing multispectral images such as smartphones and action camera, the main difference is related to operation in the field of a crime scene.

Minor issues:

1) What are the specific contributions of this paper? It is suggested to add at the end of the introduction:

Following the reviewer’s suggestions, we have modified the end of the introduction section, providing the reader with the paper motivation and the problem to be solved. We have accordingly modified the abstract to clearly show the paper contributions: “obtaining latent forensic traces from a crime scene using a modified digital camera attached to the adequate optic lens and wavelength filters with the suitable illumination to capture an image where the traces are shown in the best possible way”. 

2) There are some grammatical errors in the article, which need further careful proofreading. For example, some images are not clear enough. What is the ordinate of Figure 4?

We have proofread the paper again. In Figure 4, we have added the legends text (Wavelength and % Transmission) in the 3 subplots.

Once again, we would like to express our gratitude for the relevant effort done reviewing this paper.

Round 2

Reviewer 2 Report

Thank the authors for reply. I have no other questions.